Almost all known sauropod necks are incomplete and distorted

http://orcid.org/0000-0002-1003-5675 Taylor Michael P. dino@miketaylor.org.uk.
Department of Earth Sciences, University of Bristol , Bristol , United Kingdom
Hedrick Brandon
Electronic publication date: 2022 Jan 24
Publication date: 2022
Volume: 10
Electronic Location ID: e12810
Received 2021 Jul 14; Accepted 2021 Dec 28
Copyright: © 2022 Taylor
Copyright year: 2022
Copyright holder: Taylor
License: This is an open access article distributed under the terms of the Creative Commons Attribution License, which permits unrestricted use, distribution, reproduction and adaptation in any medium and for any purpose provided that it is properly attributed. For attribution, the original author(s), title, publication source (PeerJ) and either DOI or URL of the article must be cited.
License URL: https://creativecommons.org/licenses/by/4.0/

Keywords: Sauropod, Dinosaur, Neck, Cervical vertebrae, Preservation, Distortion, Cervicodorsal transition

Funding: The author received no funding for this work.

==============================
Sauropods are familiar dinosaurs, immediately recognisable by their great size and long necks. However, their necks are much less well known than is often assumed. Surprisingly few complete necks have been described in the literature, and even important specimens such as the Carnegie Diplodocus and Apatosaurus, and the giant Berlin brachiosaur, in fact have imperfectly known necks. In older specimens, missing bone is often difficult to spot due to over-enthusiastic restoration. Worse still, even those vertebrae that are complete are often badly distorted—for example, in consecutive cervicals of the Carnegie Diplodocus CM 84, the aspect ratio of the posterior articular facet of the centrum varies so dramatically that C14 appears 35% broader proportionally than C13. And even in specimens where the cervicodorsal sequence is preserved, it is often difficult or impossible to confidently identify which vertebra is the first dorsal. Widespread incompleteness and distortion are both inevitable due to sauropod anatomy: large size made it almost impossible for whole individuals to be preserved because sediment cannot be deposited quickly enough to cover a giant carcass on land; and distortion of presacral vertebrae is common due to their lightweight hollow construction. This ubiquitous incompleteness and unpredictable distortion compromise attempts to mechanically analyze necks, for example to determine habitual neck posture and range of motion by modelling articulations between vertebrae.

Introduction

In a paper on how the long necks of sauropods did not evolve primarily due to sexual selection (Taylor et al., 2011), one of the ideas we discussed is that sexual dimorphism between the necks of male and female sauropods, expressed as a ratio of neck lengths to shoulder height, might be an indicator of sexual selection. Rather despairingly, we wrote (Taylor et al., 2011: 4): “Available samples of sauropod taxa are unfortunately not large enough to demonstrate bimodal distribution of morphological features within any sauropod species.”

Sauropod specimens are rarely found in a form complete enough to allow even relatively rudimentary measurements to be made—for example, neck length or shoulder height. In fact, the problem is more significant than is generally realised. It is not just that we do not have large populations of well-preserved sauropod individuals, capable of being subject to statistical analyses; even individual complete sauropods are extremely rare. This is true especially of the necks, which are composed of large, fragile vertebrae that are prone to disarticulation and distortion.

The consequent incompleteness of known sauropod necks, and the ubiquitous distortion of the elements that are available, has negative consequences for taxonomic work (since we are frequently unable to compare overlapping elements of different specimens), phylogenetic analysis (due to loss of character data), developmental studies (as we are frequently unable to determine for example how pneumatic features vary along the neck in patterns mimicking ontogeny), biomechanical function (since we do not have sufficiently precise joint anatomy to accurately model neutral posture or range of motion) and ecological impact (using neck length as a proxy for feeding behaviour). While these problems are appreciated by seasoned campaigners, they are less well understood by newcomers to the field and to those whose specialisms are in adjacent fields such as extant animal anatomy, especially as most people’s initial impressions of sauropod are formed by large and apparently complete mounted specimens in public galleries.

In this paper I will first show that even the best-preserved and best-known sauropod specimens mostly have necks that are incomplete, then show that distortion of what cervical vertebrae we do have is ubiquitous and unpredictable, and finally explore the implications of this on what we can know of how these necks behaved in life.

Incompleteness

A truly complete neck would consist of all vertebrae, each of them individually complete. Unfortunately, it is rarely possible to tell from published descriptions whether a given neck is complete in this sense: necks are sometimes described superficially as “complete” when all that is meant is that some portion of each vertebra is preserved. In the cases of necks that are described in detail, it is almost always apparent that a “complete” neck is complete only in this limited sense: for example, in the Cathetosaurus lewisi holotype BYU 9047, McIntosh et al. (1996: 76) note that although all 12 cervicals are present, “10–12, particularly 12, have suffered such severe damage that it is impossible to restore them”. For the purposes of this paper then, we use “complete” in the unsatisfactory sense that at least a good part of each vertebra is present.

The determination of completeness of necks is also hindered by the problem that for many species we do not know how many vertebrae would constitute a complete neck. When necks are not found in articulation but a probably complete set of cervicals is found scattered, the possibility that additional vertebrae were present in life cannot be discounted. The mode of preservation can vary wildly with disarticulated necks, giving us a greater or lesser degree of confidence that all vertebrae are present. At one end of the spectrum, vertebrae may be scattered around, and intermixed with those of other species or even other individuals of the species, as with the numerous cervical vertebrae included in the Camarasaurus supremus complex AMNH 5761/5761a and assigned somewhat arbitrarily to three sequences (Osborn & Mook, 1921: plates LXVII–LXIX). In happier cases, such as that of the Qijianglong guokr holotype QJGPM 1001, partial sequences are articulated (in this case C2–C11) and other cervicals closely associated. However, even a neck found in articulation may be impossible to evaluate as to its completeness, as with Lavocatisaurus agrioensis (see below).

With these caveats in mind, I now survey the known complete and nearly-complete sauropod necks.

Catalogue of complete necks

Unambiguously complete necks are known from published accounts of only a few sauropod specimens. In chronological order of description, the following specimens were found with their necks complete and articulated, and have been adequately described—each of them known to include the posteriormost cervicals because the vertebral column is articulated through into the dorsal region: CM 11338, a referred specimen of Camarasaurus lentus described by Gilmore (1925). This is a juvenile specimen, and thus does not fully represent the adult morphology. (McIntosh et al., 1996: 76 claim that this specimen is the holotype, but this is not correct: YPM 1910 is the holotype—see below.)

CM 3018, the holotype of Apatosaurus louisae, described by Gilmore (1936). The neck was separated from the torso but articulated from C1–C15, though the last three cervicals were badly crushed: see below for details.

CCG V 20401, the Mamenchisaurus hochuanensis holotype, described by Young & Zhao (1972). Each vertebra is broken in half at mid-length, with the posterior part of each adhering to the anterior part of its successor; and all the vertebrae are badly crushed in an oblique plane.

ZDM T5402, a Shunosaurus lii referred specimen, described in Chinese by Zhang (1988), with English figure captions. Their figure 22 depicts the atlas. Unlike the holotype T5401, this specimen is mature.

BYU 9047, the Cathetosaurus lewisi holotype, described by Jensen (1988). (Jensen incorrectly gives the specimen number as BYU 974.) This specimen was redescribed, and the species referred to Camarasaurus, by McIntosh et al. (1996). Although all 12 cervicals are present, “10–12, particularly 12, have suffered such severe damage that it is impossible to restore them” (McIntosh et al., 1996: 76).

MACN-N 15, the holotype of Amargasaurus cazaui, described by Salgado & Bonaparte (1991) who described “22 presacral vertebrae articulated with each other and attached to the skull and sacrum, relatively complete” (Salgado & Bonaparte, 1991: 335, translated).

ZDM 0083, the holotype of Mamenchisaurus youngi, described in Chinese by Ouyang & Ye (2002) with English figure captions. Their figure 14 depicts the atlas and axis.

MUCPv-323, the holotype of Futalognkosaurus dukei, initially described by Calvo et al. (2007a) and redescribed by Calvo et al. (2007b). The neck was found in two articulated sections which fit together without needing additional vertebrae in between (Jorge O. Calvo, 2021, personal communication).

SSV12001, the holotype of Xinjiangtitan shanshanesis, described by Zhang et al. (2018). The original description of this specimen by Wu et al. (2013) included only the last two cervicals, which were the only ones that had been excavated at that time.

A few additional specimens are known to have complete and articulated necks, but have not yet been described: USNM 13786, a referred subadult specimen of Camarasaurus lentus recently mounted at the Smithsonian. The specimen “was almost completely buried before the sinews had allowed the bones to separate” (letter from Earl Douglass to William J. Holland, 22 August 1918), and photographs kindly supplied by Andrew Moore show that the atlas was preserved.

MNBH TIG3, the holotype of Jobaria tiguidensis. Sereno et al. (1999: 1343) assert that this species has 12 cervicals in all and say “One articulated neck was preserved in a fully dorsiflexed, C-shaped posture”. Paul C. Sereno (2021) personal communication, confirms that the articulated neck is MNBH TIG3.

SMA 002, referred to Camarasaurus sp. Tschopp et al. (2016), in a description of its feet, say that this specimen “lacks only the vomers, the splenial bones, the distal end of the tail, and one terminal phalanx of the right pes. The bones are preserved in three dimensions and in almost perfect articulation”.

MAU-Pv-LI-595, the “La Invernada” Titanosaur. Filippi et al. (2016) give a very brief account in an abstract. Filippi (2021) personal communication, says that the entire preserved specimen was articulated.

MAU-Pv-AC-01, an unnamed titanosaur mentioned in abstracts by Calvo, Coria & Salgado (1997) and Coria & Salgado (1999). The specimen was found in perfect articulation from skull down to the last caudal vertebrae (Rodolfo A. Coria, 2021, personal communication).

The first cervical (the atlas) in sauropods is very different in form from the other vertebrae, and small and fragile. Consequently it is easily lost. Some further specimens have necks that are complete and articulated from C2 (the axis) backwards: MB.R.4886, the holotype of Dicraeosaurus hansemanni, described by Janensch (1929), has a neck that is complete and well preserved from C2 to C12 (the last cervical). Janensch referred to this as “specimen m” and writes “It was found articulated from the 19th caudal vertebra to the 9th cervical vertebra inclusive. The proximal part of the neck from the 8th cervical vertebra up to the axis was bent ventrally and lay at right angles to the distal part of the neck.” (Janensch, 1929: 41).

PMU 233, the holotype of Euhelopus zdanskyi, described by Wiman (1929) as “exemplar a” and redescribed by Wilson & Upchurch (2009).

ZDM T5401, the subadult holotype of Shunosaurus lii, described in Chinese by Zhang, Yang & Peng (1984). The quarry map (Zhang, Yang & Peng, 1984: figure 1) suggests that the atlas is missing.

MCT 1487-R, informally known as “DGM Series A”, described by Powell (2003). Gomani (2005: 9) summarises as “12 cervical vertebrae, except the atlas, preserved in articulation with three proximal dorsal vertebrae”.

GCP-CV-4229, the holotype of Spinophorosaurus nigerensis, described by Remes et al. (2009). This species is known from two specimens, of which the holotype was found in very good condition and well articulated from C2 to C13, the last cervical (Fig. 1). The atlas seems to be missing (Kristian Remes, 2021, personal communication; Ralf Kosma, 2021, personal communication).

Figure 1 Spinophorosaurus nigerensis holotype GCP-CV-4229 in situ during excavation in the region of Aderbissinat, Thirozerine Dept., Agadez Region, Republic of Niger.

Reproduced from Remes et al. (2009: Figure 1).

One other sauropod is complete from the first cervical, but probably not to the last: MOZ-Pv1232, the holotype of Lavocatisaurus agrioensis, described by Canudo et al. (2018). This is complete from C1–C11. Canudo’s guess is that this is the complete neck (Jose I. Canudo, 2021, personal communication), but the specimen doesn’t demand that conclusion and no known eusauropod is definitively known to have fewer than 12 cervicals. However, Upchurch (2021) personal communication, notes that the dicraeosaurid neosauropod Amargasaurus has 22 presacral vertebrae in total, and difficulties in locating the cervicodorsal junction (see below) make it impossible to state confidently whether it had 12 or only 11 cervicals, so caution is warranted here.

Other sauropod specimens have necks that are complete and articulated from further back in the cervical sequence: YPM 1910, the holotype of Camarasaurus lentus, a mounted specimen described by Lull (1930). The neck is complete from C2 or C3, Lull was uncertain which. (It should be possible to clear that up by re-examining the specimen in light of what has subsequently become known about Camarasaurus lentus, but this work has not yet been done.)

SMA 0004, the holotype of Kaatedocus siberi, described by Tschopp & Mateus (2012). Cervicals 3–14 are preserved.

AODF 888 (informally “Judy”), probably referrable to Diamantinasaurus, briefly described by Poropat et al. (2019). Preserved from C3 or maybe C4. “One posterior cervical (XIII or XIV) found several metres from articulated series, but appears to slot nicely into the gap between the articulated cervical series and the unprepared thoracic section, which might include at least one additional cervical (XIV or XV)” (S. F. Poropat, 2021, personal communication).

Several necks are probably nearly complete, but it is not possible to know due to their not being found in articulation: CM 84, the holotype of Diplodocus carnegii, described by Hatcher (1901). C2–C15 are preserved, though not all in articulation; C11 may be an intrusion: see below for details.

ZDM T5701, the holotype of Omeisaurus tianfuensis, described by He, Li & Cai (1988). The neck was not articulated (He, Li & Cai, 1988: figure 1), and was missing “two elements or so” (He, Li & Cai, 1988: 120).

QJGPM 1001, the holotype of Qijianglong guokr, described by Xing et al. (2015). On page 8, the authors say “The axis to the 11th cervical vertebra were fully articulated in the quarry. The atlas intercentrum and the 12th–17th cervical vertebrae were closely associated with the series.”

MNBH TIG9, a referred specimen of Jobaria tiguidensis. Wilson (2012: 103) writes that this specimen “includes a partially articulated series of 19 vertebrae starting from the axis and extending through the mid-dorsal vertebrae.”

MNBH TIG6, another referred specimen of Jobaria tiguidensis, which has not been mentioned in the literature. Paul C. Sereno (2021) personal communication, says that it is “a subadult partial skeleton with excellent neck” and that “the sequence was articulated from C2–C11. Most of the ribs were attached as well.”

Note. The Jobaria tiguidensis individuals previously had specimen numbers beginning MNN, but the Musée National du Niger changed its name to Musée National Boubou Hama and the specimen numbers have changed with it.

The breakdown of these complete and near-complete necks is interesting (Fig. 2). Non-neosauropods are relatively well represented, both inside and outside of Mamenchisauridae—although it is unfortunate many of these specimens are not well described in English: two of the ten are of Jobaria, for which the cursory summary of Sereno et al. (1999) remains the only published description, and some of the Chinese sauropods are described only in Chinese.

Figure 2 Simplified phylogeny of Sauropoda, showing counts of complete and near-complete necks.

Captions: C, complete and described; U, complete but undescribed; –1, missing the atlas but otherwise complete; O, other near-complete necks (see text for details); T, total.

Diplodocoids are surprisingly poorly represented, with only a single specimen in each of Dicraeosauridae and Diplodocidae that is complete. Brachiosaurids have absolutely no representation—see below on how unconvincing the neck of Giraffatitan is. More advanced titanosauriforms are better represented, but there is still only one with a complete neck, Futalognkosaurus dukei. By contrast, the single genus Camarasaurus is very well represented, with five specimens of which four are fully complete (though only two of those have been described). Probably this does not indicate a taxon-specific taphonomic signal, but follows from the sheer abundance of Camarasaurus specimens—an abundance likely influenced by over-lumping of multiple rather different species into a single genus.

It is surprising, though, that the second and third best represented sauropods in museums, Diplodocus and Apatosaurus, are both barely represented in terms of complete necks. And while the number of complete and nearly-complete necks among somphospondyls, including titanosaurs, is encouraging, it is disappointing that so many of them are not yet described.

At the time of writing (21 December 2021), the Paleobiology Database (https://paleobiodb.org/) lists 342 sauropod species. The nine unambiguously complete and articulated necks therefore represent only one in 38 known sauropod species—and recall that even these are mostly “complete” only in the weak sense of preserving some part of each cervical vertebra.

As best we can tell, only one sauropod species, Camarasaurus lentus, is known from more than a single complete neck. Of the two individuals, CM 11338 is a juvenile and USNM 13786 is a subadult, so the mature morphology is unknown. If we allow necks missing the atlas, then there are also two individuals of Shunosaurus lii: ZDM T5401, the subadult holotype, which is missing its atlas; and ZDM T5402, an adult referred specimen whose neck is complete. (These specimens have not been described in English.) With sample sizes this small, it is not possible even in principle to determine whether there is a bimodal distribution in the length of any sauropod’s neck.

Several well-known sauropod specimens are often thought of as having complete, undamaged necks, but in each case the truth is less clear. I now discuss three important specimens.

Diplodocus carnegii CM 84

The Carnegie Diplodocus is one of the most recognised dinosaurs in the world: not only is the original specimen, CM 84, on display as a mounted skeleton in the Carnegie Museum in Pittsburgh, but casts are displayed in many other major museums (e.g. the Natural History Museum in London, the Museum für Naturkunde Berlin and the Muséum National d’Histoire Naturelle in Paris.) The neck appears complete in these mounted skeletons, with fifteen cervical vertebrae, and is illustrated as such by Hatcher (1901: plate 8); Fig. 3. However, the situation is not as clear as it appears in these exhibits.

Figure 3 Neck of Diplodocus carnegii holotype CM 84, as reconstructed by Hatcher (1901: plate XIII), with 15 undamaged cervical vertebrae.

Holland (1900: 816), in the first published account of the Carnegie Diplodocus, assigned to this specimen only eleven cervicals, noting (on p. 817) that: The cervicals were for the most part interarticulated, all lying in such position as to show the serial order […] Eleven are found in the specimen at the Carnegie Museum, atlas and axis being as yet undiscovered.

Allowing for the missing atlas and axis, Holland concluded only that the cervical count was “at least 13”.

However, Hatcher (1900: 828–829) corrected this count later the same year: About 45 feet (14 m) of the vertebral column is preserved in our specimen. When discovered the vertebrae did not lie in a connected and unbroken series, yet there can be little doubt that they all pertain to the same individual […] Unfortunately no diagram was made, at the time of exhuming the remains, showing the relative position of each of the vertebrae in the quarry […] Early last spring, at the request of the writer, Mr. W. H. Reed (who assisted in unearthing the skeleton), while again on the ground, made a diagram of the quarry, showing the relative positions, as he remembered them, of the various bones of the skeleton.

Despite this uncertainty, Hatcher asserted (p. 828–829): “In all 41 vertebrae are represented, including 14 cervicals (all but the atlas) […] Assuming that no vertebrae are missing from our series the vertebral formula of Diplodocus should now be written as follows: Cervicals, 15 […] The number of cervical vertebrae in Diplodocus is definitely fixed at at least 15.”

Hatcher’s (1900) paper is unsatisfactory in that it gives no reason for his revision of the cervical count. Hatcher also hedged by leaving open the possibility of there being more than 15 cervicals. The lack of a reliable quarry map is unfortunate.

In his subsequent monograph, Hatcher (1901: 4) expanded on the completeness and condition of the material as follows (emphasis added): [Diplodocus carnegii holotype CM 84] has been entirely freed from the matrix and is found to consist of [appendicular material and] forty-one vertebrae divided as follows: fourteen cervicals including the axis, eleven dorsals, four sacrals, and twelve caudals. These vertebrae are for the most part fairly complete, though unfortunately the sacrals and anterior cervicals are more or less injured. This series of forty-one vertebrae are believed to pertain to one individual and to form an unbroken series from the axis to the twelfth caudal, although as was shown in a previous paper, there is some evidence that there are perhaps one or more interruptions in the series and that one or more vertebrae are missing. On the other hand, as will appear later, it is not entirely impossible that at least one vertebra of this supposed series pertains to a second individual belonging perhaps to a distinct genus.

Hatcher (1901: 11) went on to quote a statement from A. S. Coggeshall, who had assisted in the excavation, explaining in more detail how the elements of the neck were discovered: [The] last (fifteenth) cervical was considerably removed from the succeeding dorsals and less so from the preceding cervicals. Commencing with the next vertebra (cervical fourteen), the direction of the entire cervical series was altered so that it lay with its axis almost at right angles to that of the dorsal series. The cervicals extended in an almost straight line from the fourteenth to the fifth, but there was a considerable gap between cervicals 11 and 10, while the axis and cervicals three, four and five were doubled back under the succeeding vertebrae.

This account almost explains why Holland underestimated the number of cervicals: the anteriormost four, lying under more posterior cervicals, had not yet been found. However, if ten cervicals (C6–C15) had been found and the atlas and axis were both missing, Holland (1900: 816) would surely have stated “Ten are found in the specimen at the Carnegie Museum, atlas and axis being as yet undiscovered” rather than eleven. Some mystery remains: perhaps Holland was aware of one of the anteriormost four preserved cervicals.

Coggeshall’s description is somewhat corroborated by Reed’s quarry map, which is included as Plate 1 of Hatcher’s (1901) monograph (Fig. 4). However, the map is in some respects at odds with the description: for example, it shows all 13 vertebrae C2–C14 in a single straight line rather than indicating that C2–C5 were doubled back; and it shows gaps both between C10 and C11 (as stated), and also between C11 and C12 (not mentioned in the text).

Figure 4 W. H. Reed’s diagram of Quarry C near Camp Carnegie on Sheep Creek, in Albany County, Wyoming.

The coloured bones belong to CM 84, the holotype of Diplodocus carnegii; other bones belong to other individuals, chiefly of Brontosaurus, Camarasaurus and Stegosaurus. Modified (cropped and coloured) from Hatcher (1901: plate I). Cervical vertebrae are purple (and greatly simplified in outline by Reed), dorsals are red, the sacrum is orange, caudals are yellow, limb girdle elements are blue, and limb bones are green.

Regarding the vertebra that might pertain to “a second individual belonging perhaps to a distinct genus”, Hatcher (1901: 22) explained: “Eleventh Cervical.—This vertebra is so unlike either the immediately preceding or succeeding vertebrae that if it had been found isolated it would have been unhesitatingly referred to a different genus. Mr. Coggeshall, however, assures me that it was interlocked with the succeeding, or twelfth cervical.” Yet, as noted, the quarry map suggests that there was some distance between C11 and C12, perhaps invalidating Coggeshall’s assertion. It is to be lamented that both the map and the description were created some time after the excavation actually took place, by which time memories had evidently become unreliable.

In conclusion, Diplodocus carnegii most likely had fifteen cervicals, but may have had more (if some vertebrae were not recovered), or maybe fewer (if C11 was misassigned). Furthermore, the anterior cervicals are damaged in a way that is not at all apparent from Hatcher’s drawings (plate III) or photographs (plate IV) because they were restored before these illustrations were prepared. As Hatcher (1901: 23) noted, “The work of freeing these vertebrae from the matrix and restoring them was for the most part done during my absence in the field. Unfortunately no drawings or photographs were taken prior to the process of restoring with colored plaster.” (In the early 20th Century, it was routine to restore damaged fossils in ways that completely obscured the degree of damage: see Fig. 5)

Figure 5 Three images of presacral vertebra 6 (probably dorsal 7) of Brachiosaurus altithorax holotype FMNH P25107, in right lateral view, showing misleading restoration.

Left: Field Museum photograph CSGEO16166, photographer Charles Carpenter, taken in 1905, the year after Riggs’s (1904) descriptive monograph. Note the “crazy-paving” effect of the many cracks and missing areas of bone surface. Middle: Illustration of the same vertebra in Riggs (1904: plate LXXII). Note that the damage to the vertebral surface is not depicted. Right: photograph of the same vertebra, taken by the author in 2005. Note that the damage apparent in the 1905 photograph is no longer visible: the vertebra seems to have been painted to conceal its incompleteness.

Apatosaurus louisae CM 3018

Apatosaurus louisae is the best known species of Apatosaurus, since its holotype CM 3018 is much more complete and better preserved than that of the type species A. ajax (YPM 1680), or that of the closely related Brontosaurus excelsus (YPM 1980).

The specimen was collected by Earl Douglass in 1909 and 1910, from what was then known as the Carnegie Museum Dinosaur Quarry near Jensen, Utah, and is now Dinosaur National Monument. It was mounted for exhibition in 1913, and somewhat belatedly named the type of a new species in a brief initial description by Holland (1915). He noted that “the specimen consists of a series of vertebrae, complete from the atlas to nearly the end of the tail” and appendicular material; but also that “the cervical vertebrae had been separated from the dorsals and shifted, but the entire series was found articulated in regular order” (p. 143). (Holland’s description also mentioned that “a skull, which judging by its location, belongs to the specimen, was found within eleven feet of the atlas. It does not differ greatly in form from the skull which belongs to Diplodocus”. Had Holland stuck to his guns, Apatosaurus could have been restored with its correct skull 63 years before Berman & McIntosh (1978) corrected Marsh’s long-standing and influential misapprehension that it had a Camarasaurus-like skull.)

Holland stated (p. 144) that he had “in preparation a large monographic paper relating to the genus, based in part upon [CM 3018]”. However, completion was long delayed, and Holland died in 1932 before in was ready to be published. It was eventually brought to completion by Gilmore (1936) and it is from this monograph that the species is primarily known.

Gilmore’s monograph explains that all is not as it seems in the neck of his specimen. He notes (p. 191) that “there was some distortion due to the compression to which [the cervicals] had been subject, but this has been largely corrected during preparation”—a questionable decision, as it means that the shapes of the vertebrae as originally found are now lost, and cannot be subjected to more modern retrodeformation techniques (e.g. Tschopp, Russo & Dzemski, 2013). He continues “Cervicals thirteen, fourteen, and fifteen, however, were so badly crushed that it was thought best to replace them in the mounted skeleton by plaster restorations of these vertebrae”, although he does claim that “they are, however, sufficiently well preserved so that most of their important characteristics can be determined”. The caption to Gilmore’s plate XXIV reads “Cervical vertebrae of Apatosaurus louisae. Type, No. 3018 […] Cervicals 13, 14 and 15 have been much restored from badly crushed originals, and should be used with caution.” It is also evident from this plate that most of C5 is also missing, although this is not acknowledged in the text. As noted by Upchurch (2000), the poor condition of the posterior cervical vertebrae, and their replacement by plaster models in the mounted skeleton, compromise the validity of biomechanical modelling based on this specimen, such as that of Stevens & Parrish (1999).

In conclusion, while the articulation of the cervical sequence of CM 3018 leaves little doubt that all cervicals are present and in the correct order, the crucial posterior cervicals are largely uninformative.

Giraffatitan brancai MB.R.2181

This specimen is the paralectotype of Giraffatitan brancai (= “Brachiosaurus” brancai). Much of the material is incorporated in the mounted skeleton in the atrium of the Museum für Naturkunde Berlin, which remains the largest substantially real mounted skeleton of a terrestrial animal anywhere in the world. (There are larger mounts of sauropods, such as the skeletons of Patagotitan at the AMNH and FMNH, but these are casts and sculptures, not real material.) While most of the material of the Berlin brachiosaur mount is real fossil bones, the presacral vertebrae are too heavy and fragile to mount: instead, high quality sculptures are used, and the vertebrae themselves are held in collections.

The presacrals in the mount are based on real bones that are from two specimens—the lectotype MB.R.2180 (formerly known as SI) and the paralectotype MB.R.2181 (formerly SII). The former includes cervicals 2–7, an assignment that can be accepted with some confidence if the vertebrae indeed form a sequence because the axis, C2, in sauropods is very distinctive, having a completely different anterior articular surface from all the subsequent cervicals (see e.g. Janensch, 1950: figs. 9–16, cf. figs. 17–48.) MB.R.2181 includes cervicals assigned by Janensch to positions 3–13 (although almost all of them are damaged, some very severely).

However, the two individuals MB.R.2180 and MB.R.2181 were found together in a single quarry (designated Quarry S). Bones of the two individuals were jumbled up together, with little articulation, as shown in the quarry map, redrawn by Heinrich (1999: figure 16; Fig. 6) from an original drawn in the field by Werner Janensch. Any reconstruction—or even assignment of individual vertebrae to one specimen or the other—must be considered provisional.

Figure 6 Quarry map of Tendaguru Site S, Tanzania, showing incomplete and jumbled skeletons of Giraffatitan brancai specimens MB.R.2180 (the lectotype, formerly HMN SI) and MB.R.2181 (the paralectotype, formerly HMN SII).

Cervical material is highlighted in strong yellow, while the remaining elements are desaturated. Anatomical identifications of MB.R.2181 are underlined. Elements of MB.R.2180 could not be identified on the map with certainty. Modified from Heinrich (1999: Figure 16), which was itself redrawn by Heinrich from an original field sketch by Werner Janensch.

I have previously suggested (Taylor, 2009: 800–801) that the distinctively high-spined dorsal vertebra usually considered the fourth of MB.R.2181 may not actually belong to that specimen, or even that taxon. Instead, this unusually tall vertebra may belong to an animal more closely resembling the Tendaguru titanosauriform briefly described by Migeod (1931) and which I am in the process of redescribing (Taylor, 2005, Taylor in prep at https://github.com/MikeTaylor/palaeo-archbishop/). If this vertebra is indeed not part of MB.R.2181 then the most likely inference is that it is part of MB.R.2180. This would be unfortunate if these two specimens were indeed representatives of different taxa. The smaller and less complete MB.R.2180, rather than the larger, more complete and better known MB.R.2181, is the lectotype (Janensch, 1935). Therefore, the ICZN rules dictate that the name Giraffatitan brancai would adhere to MB.R.2180, and that a new name would be required for the better-known MB.R.2181. Since this species was thought until relatively recently to be a species of the North American genus Brachiosaurus (see Taylor, 2009), a further reassignment would mean that this charismatic and iconic specimen would become known by a third different name in little more than a decade. To avoid this outcome, an ICZN petition may be warranted.

Janensch (1950: 33) indicates that the confusion of the cervical vertebrae is not as bad as that of the dorsals, but the situation is still far from clear, as Janensch’s description explains: The vertebrae from the 3rd to 15th presacrals [of MB.R.2181] lay in articulation in a consolidated lime sandstone lens; of them, the 3rd to 5th vertebrae are tolerably complete, the remaining 10 vertebrae were articulated with one another, with one interruption that arose when the 8th presacral vertebra rotated out of the series and was displaced. [Translation by Gerhard Maier.]

Janensch’s words are somewhat at odds with the quarry map, which shows cervical material scattered around the northeastern part of the quarry (Fig. 6). Other quarry sketches drawn by Janensch, including one reproduced by Heinrich (1999: figure 18), certainly show at least some cervicals having been found in articulation, but it is not clear what the correspondence is between the field numbers on these sketches and subsequently assigned element identifications.

So it is possible, though unlikely, that there might have been other displaced cervicals, before or after the one designated “8th”, that were not recovered. Neither can we be wholly confident that the anteriormost preserved cervical in the MB.R.2181 series is really C3. Its identification is based on the overlap with vertebrae of MB.R.2180, but we cannot be certain that MB.R.2180 is a member of the same species as MB.R.2181. Perhaps the anteriormost preserved cervical is really C4? Or perhaps some of the “MB.R.2181” cervicals really belong to MB.R.2180. This is not particularly likely, as Janensch pointed out, due to the difference in size between the two specimens—the MB.R.2181 cervical centra are about 30–40% longer than those assigned the same serial position in MB.R.2180. But the size difference cannot be considered a conclusive argument, especially given the uncertainty in serial-position assignments in both specimens.

In conclusion: Giraffatitan brancai probably had thirteen cervicals, but may have had more, or possibly less; and the neural arches are only known for cervicals 3, 4, 5 and 8 in MB.R.2181 (if these are the correct serial positions for those vertebrae). If MB.R.2180 is indeed a member of the same species then cervicals 2–7 are known from well-preserved elements, but no more. All of this uncertainty is exacerbated by the problem that no complete or even nearly complete neck of any other brachiosaur has been described.

To summarise this section, not only are complete sauropod necks in very short supply, even those that are considered complete cannot generally be confidently considered so, and complexities of interpretation bedevil the best-known specimens.

Distortion

Even in necks where most or all of the vertebrae are present and largely complete, extensive distortion is common. This is difficult to quantify, even in principle, given the very complex shapes of sauropod cervicals. But we can take tentative steps towards recognising the extent of the problem by considering one simple aspect: the shapes of the cotyles of consecutive vertebrae.

In sauropod cervical vertebrae (and most dorsal vertebrae), the posterior articular face of the centrum is called the cotyle, due to its distinctive hollow shape. The anterior articular face is convex, and so is called the condyle. The cotyle of one vertebra and the condyle of the succeeding one form a ball-and-socket joint (see Taylor & Wedel, 2013b: figures 2–3), similar to the condition in extant horses and camels (see Taylor & Wedel, 2013b: figures 20–21) among other animals.

In extant animals, including birds, crocodilians, lizards and mammals, the articular facets of consecutive vertebrae are of much the same shape, varying only gradually along the neck. In particular, the aspect ratio of the facet—its width:height ratio—remains constant or nearly so (Figs. 7–10). However, in the fossilised necks of sauropods, it is not unusual for even consecutive vertebrae to be crushed in opposite directions, giving their cotyles (apparently) wildly different aspect ratios.

Figure 7 Cervical vertebrae 5–11 of an ostrich, Struthio camelus, in posterior view, showing that articular facet shape remains similar along the column.

Specimen kindly provided by Matt Cobley.

Figure 8 Cervical and dorsal vertebrae (C5–8 and D1–2) of a juvenile alligator, Alligator mississippiensis, in anterior view, showing that articular facet shape remains similar along the column.

Specimen kindly provided by Matthew J. Wedel.

Figure 9 Sequences of cervical vertebrae of extant animals, showing that articular facet shape remains similar along the column.

Top. Cervical vertebrae 3–7 of a mature savannah monitor lizard, Varanus exanthematicus, in anterior view. (The cervicals of monitor lizards, unlike those of sauropods and most mammals, are procoelous, with the anterior facet being concave and the posterior convex.) Bottom. cervical vertebrae 2–5 of a mature house-cat, Felis catus, in posterior view.

Figure 10 Cervical vertebrae of a baby giraffe, Giraffa camelopardalis, in posterior view, showing that articular facet shape remains similar along the column.

Top row, left to right: cervicals 7, 6 and 5; bottom row, left to right: cervicals 4, 3 and 2. Despite changes in the vertebrae along the column, the flattened inverted pentagon shape of the articular facets remains similar along the sequence. (Note that extensive cartilage caps existed on the articular facets of this very young specimen, but were lost in preparation.)

Consider for example the Giraffatitan brancai lectotype MB.R.2180 (formerly HMN SI), one of the best preserved sauropod neck series. Cervicals 4 and 6 of this specimen are shown in posterior view in Fig. 11. (The intermediate cervical 5 has part of its cotyle rim broken off, and cannot be reliably measured.) Measuring from the photos, the width:height ratio of C4 (on the left) is 683/722 pixels = 0.95, and that of C6 (on the right) is 1,190/820 pixels = 1.45. So these two vertebrae—from the same neck, and with only one other vertebrae coming in between them—differ in preserved cotyle aspect ratio by a factor of 1.53.

Figure 11 Cervical vertebrae 4 (left) and 6 (right) of Giraffatitan brancai lectotype MB.R.2180 (formerly HMN SI), in posterior view.

Note the dramatically different aspect ratios of their cotyles, indicating that extensive and unpredictable crushing has taken place. Photographs by the author.

As a second example, consider the single most studied sauropod neck specimen in the world, that of the Diplodocus carnegii holotype CM 84. Figure 12 shows adjacent cervicals 13 and 14, in posterior view. Note that the posterior part of the neck was considered well preserved by Hatcher (1901), with only anterior vertebrae noted as having been damaged. Measuring from Hatcher’s photos, the width:height ratio for C14 (on the left) is 342/245 pixels = 1.40. For C13 (on the right), it is 264/256 pixels = 1.03. So C14 is apparently 35% broader than its immediate predecessor.

Figure 12 Cervical vertebrae 14 (left) and 13 (right) of Diplodocus carnegii holotype CM 84, in posterior view.

Note the dramatically different aspect ratios of their cotyles, indicating that extensive and unpredictable crushing has taken place. Modified from Hatcher (1901: plate VI).

There is no established metric for quantifying change in a measure, such as the aspect ratio of articular surfaces, along a vertebral column. Table 1 offers two candidate metrics and shows how they are worked out for six of the seven specimens discussed in this section. (The cat has too few available vertebrae for the metric to be meaningful.) Using the first metric, average difference between aspect ratios in consecutive vertebrae, the young juvenile giraffe, the monitor lizard, the alligator and the ostrich all score in the range 4.3–6.3, while the two sauropods score 9.1 and 16.3. Using the second metric, average ratio between aspect ratios of consecutive vertebrae, the four extant animals score in the range 7.0–8.9, and the sauropods 12.0 and 22.1. It would be useful in future to calculate these metrics for larger sets of extant and fossil vertebrates, and to see whether it is generally the case that the variation metrics are higher for fossils than for extant vertebrates. For now, though, this tentative initial analysis corroborates the “eye-test” conclusion that variation is significantly greater along sauropod necks.

Table 1 Serial variation in preserved aspect ratios of consecutive cervical vertebrae in extant animals, from several major amniote clades, and sauropods.

Mammalia: giraffe Giraffa camelopardalis, young juvenile specimen, C2–C7. Lacertilia: savannah monitor, Varanus exanthematicus, C3–C7. Crocodylia: alligator Alligator mississippiensis, juvenile specimen, C5–D2; Aves: ostrich Struthio camelus. C5–C11. Sauropoda: Diplodocus carnegii holotype CM84 and Giraffatitan brancai lectotype MN.R.2180 (formerly HMN SI). Vertical and horizontal columns contain measurements of the posterior articular surfaces of the vertebrae, except for the procoelous alligator vertebrae for which anterior articular surface was used. For the saddle-shaped ostrich cervicals, the maximum height and width of the posterior articular surface was used, rather than the shorter midline distances. Measurements are in arbitrary units (e.g. mm or pixels in a photograph) but consistent within any one specimen. V/H is the aspect ratio of the measured surface, the ratio of the vertical and horizontal measurements, so that values less than 1.0 indicate a surface wider than tall, and values greater than 1.0 indicate a surface taller than wide. Note that this ratio is independent of the measurement units. 1st diff measures the absolute value of the difference between the V/H ratios of each vertebra and its successor. Max diff is the highest value of 1st diff for each specimen, and Avg diff is the average value; ×100 is this value times 100, a measure of the serial variation along the column. 1st ratio is the “absolute ratio” of the V/H values of one vertebra and its successor, normalized so that when the ratio is less than 1.0 its inverse is used. Max ratio is the highest value of 1st ratio for each specimen, and Avg ratio is the average value; –1 × 100 is this value minus one, multiplied by 100, another measure of the serial variation along the column. Note that by both measures, the variation is articular surface aspect ratios is greater in the two sauropods than in any of the extant animals, even though these are among the best preserved sauropod necks.

	Vertebra	Vertical	Horizontal	V/H	1st diff	Max diff	Avg diff	×100	1st ratio	Max ratio	Avg ratio	–1 × 100	
Giraffe	
	C2	437	638	0.685	0.086	0.151	0.062	6.201	1.126	1.208	1.086	8.636	
	C3	499	647	0.771	0.014				1.018				
	C4	471	600	0.785	0.014				1.018				
	C5	513	665	0.771	0.045				1.062				
	C6	531	731	0.726	0.151				1.208				
	C7	510	581	0.878									
Monitor	
	C3	155	255	0.608	0.020	0.081	0.043	4.270	1.034	1.137	1.071	7.113	
	C4	181	308	0.588	0.007				1.012				
	C5	179	301	0.595	0.081				1.137				
	C6	194	287	0.676	0.062				1.102				
	C7	189	308	0.614									
Alligator	
	C5	202	217	0.931	0.060	0.121	0.055	5.546	1.068	1.159	1.070	6.986	
	C6	203	233	0.871	0.016				1.019				
	C7	213	240	0.888	0.121				1.159				
	C8	203	265	0.766	0.073				1.095				
	D1	198	236	0.839	0.007				1.008				
	D2	203	244	0.832									
Ostrich (Measuring maximum distances)	
	C5	156	207	0.754	0.036	0.092	0.063	6.298	1.048	1.135	1.089	8.903	
	C6	173	219	0.790	0.027				1.035				
	C7	177	232	0.763	0.084				1.123				
	C8	161	237	0.679	0.092				1.135				
	C9	172	223	0.771	0.048				1.066				
	C10	170	235	0.723	0.091				1.126				
	C11	180	221	0.814									
Diplodocus (CM 82)	
	C2	33	36	0.917	0.087	0.243	0.091	9.130	1.105	1.139	1.120	11.955	
	C3	39	47	0.830	0.116				1.139				
	C4	52	55	0.945	0.107				1.127				
	C5	52	62	0.839	0.045				1.054				
	C6	61	69	0.884	0.009				1.010				
	C7	70	80	0.875	0.032				1.036				
	C8	68	75	0.907	0.032				1.036				
	C9	91	104	0.875	0.061				1.069				
	C10	102	109	0.936	0.123				1.152				
	C11	117	144	0.813	0.056				1.069				
	C12	139	160	0.869	0.048				1.055				
	C13	143	156	0.917	0.243				1.361				
	C14	134	199	0.673	0.228				1.339				
	C15	156	173	0.902									
Giraffatitan (MB.R.2180)	
	C2	864	717	1.205	0.180	0.378	0.163	16.288	1.176	1.568	1.221	22.100	
	C3	695	678	1.025	0.018				1.018				
	C4	672	644	1.043	0.378				1.568				
	C5	675	1,014	0.666	0.031				1.047				
	C6	828	1,188	0.697	0.207				1.297				
	C7	640	708	0.904									

It might be argued that variation in cotyle shape in sauropod necks arises from mechanical factors. Since their necks were elongate, segmented cantilevers, they become broader towards their base, and this could be reflected in cotyle shape. However, we would expect mechanical influences such as this to produce gradual monotonic variation—cotyles either becoming consistently broader or consistently taller towards the base of the neck. But this is not what we observe: instead, the preserved aspect ratios of sauropod condyles vary erratically along the neck. Such extreme variation in apparent aspect ratio of the cotyles of adjacent and near-adjacent cervical vertebrae can only be the result of extensive and unpredictable crushing.

This variation in preservation of aspect ratios has implications for calculating the elongation index (EI) of vertebrae, a calculated character widely used in phylogenetic analyses and elsewhere. For example, Janensch’s (1950: 39) table of measurements for the Giraffatitan brancai lectotype MB.R.2180 (formerly HMN SI) gives the centrum length and cotyle height of C4 as 45.7 and 13.8 cm, for an EI sensu Wedel, Cifelli & Sanders (2000: 346) of 3.31. For C6, he gives centrum length and cotyle height as 69.1 and 15.0, for a much greater EI of 4.6. But if the true cotyle proportions of C6 were the same as those of C4, then the cotyle height corresponding to its width of 22.1 would be a much taller 23.1 cm, yielding an EI of only 3.0. Some recent work uses the aEI of Chure et al. (2010), in which the centrum length is divided not by cotyle height but by the average of cotyle height and width. This goes some way towards mitigating the confounding effects of crushing, but cannot fully allow for its unpredictable effects: for example, oblique crushing may increase both maximum height and width of the cotyle, yielding a misleadingly low aEI. At the very least, we need to be circumspect in our use of elongation indices as phylogenetic characters or ecological correlates.

Deformation of the articular cotyle is only one example of the many ways in which sauropod vertebrae, with their complex and fragile anatomy, are subject to crushing, It is certain that other parts of the vertebrae, especially the delicate lateral processes and zygapophyseal rami, were also often distorted, and it is generally not possible to reliably restore them to their undistorted state.

In addition to such distortions of individual processes of vertebrae, systematic distortion of entire vertebrae is common. For example, in CCG V 20401, the holotype specimen of Mamenchisaurus hochuanensis, for which the neck is complete and articulated, every vertebra of the neck and trunk is sheared and rotated such that the left side is displaced downwards (Taylor, 2004, personal observation of mounted casts in Copenhagen, Denmark; Trzic, Slovenia; and Chicago, USA). This distortion is illustrated for dorsal vertebra 2 by Young & Zhao (1972: figure 6); although they do not illustrate it for a cervical vertebra, it is present throughout the column.

While the Mamenchisaurus hochuanensis distortion is very apparent, subtler distortions are ubiquitous but easier to overlook. For example, the Giraffatitan brancai holotype cervical vertebrae MB.R.2180 (formerly HMN SI) appear undistorted to the naked eye, but manual articulation of C2 and C3 demonstrates that sufficient distortion has occurred to prevent the bones being posed in ways that the live animal surely could have achieved (Fig. 13).

Figure 13 Manipulation of consecutive sauropod vertebrae by hand. Cervicals 2 and 3 of Giraffatitan brancai lectotype MB.R.2181 (formerly HMN SI).

I attempted to articulate these two vertebrae, and empirically determine the feasible range of motion. Due to subtle distortion of the zygapophyses of these vertebrae, it was not possible to articulate C2 in a more extended position relative to C3 than shown here. Photograph by Michael P Taylor.

Location of cervicodorsal junction

One further issue impedes our ability to analyse and compare the necks of different sauropods, and that is the difficulty of identifying the last cervical vertebra—and therefore of defining how many vertebrae make up the neck (and how many make up the dorsal series). In general it is easy to tell cervical and dorsal vertebrae apart: for example, compare C13 and D3 of the Diplodocus carnegii holotype CM 84 (Hatcher, 1901: plates III and VII). The cervical vertebra is relatively low, its centrum is elongated, its neural spine is roughly triangular and its parapophysis hangs down below the centrum and has a cervical rib fused to it and the diapophysis (though this latter character is not consistent in sauropods: see below). By contrast, the dorsal vertebra is tall, its centrum is short, its neural spine is anteroposteriorly compressed, its parapophysis is up on the dorsal half of the centrum, and no rib is fused to it. But the change in these characters is gradual, and at the transition it is much more difficult to distinguish between cervical and dorsal vertebrae. Compare C15 and D1 of CM 84 (Hatcher, 1901: plates III and VII once more). Had Hatcher classified his C15 as the first dorsal, or his D1 as the last cervical, it would not be obvious that this was incorrect. Similarly, Wilson & Upchurch (2009: 20) found it difficult to unambiguously identify the first dorsal vertebra even when dealing with the relatively complete and well-preserved presacral sequence of Euhelpus zdanskyi.

The difficulty of locating the cervicodorsal junction is exacerbated by the lack of a single standard definition. Several exist: Rib fusion. Hatcher (1901: 25) writes “The dorsals are distinguished from the cervicals by supporting free instead of fixed ribs …”

Ventral centrum shape. Hatcher (1901: 25–26) continues “… and in having the inferior surface of the centra regularly convex transversely instead of concave in either direction”.

Scapular support. In his description of Haplocanthosaurus two years later, Hatcher (1903: 8) uses a completely different definition: “That [D1] was a dorsal is conclusively shown not by the presence of tubercular and capitular rib facets showing that it supported on either side a free rib […] The character in this vertebra distinguishing it as a dorsal is the broadly expanded external border of the anterior branch of the horizontal lamina [i.e. what we would now call the prezygadiapophyseal lamina …] to give greater surface for the attachment of the powerful muscles necessary for the support of the scapula.” (See Fig. 14)

Parapophysis location. Hatcher (1901: 16) notes “In the fifth dorsal the capitular facet is on the middle of the neural arch, while in dorsals four and three it has shifted down to the centrum and encroached upon the pleurocentral cavities of these vertebrae. In dorsals two and one it lies wholly inferior to that cavity”, and his illustrations show that it is well below the centrum in all cervical vertebrae.

Gene expression. Wilson (2002: 226) notes that “the cervicodorsal transition in many tetrapods, for instance, appears to be defined by the expression boundary of the Hoxc-6 gene”.

Figure 14 Cervicodorsal transition of Haplocanthosaurus priscus holotype CM 572.

Top row, right lateral view; bottom row, anterior view. From left to right, D1, C15 and C14. Pink highlights indicate the expanded lateral surface of the prezygadiapophyseal lamina, anchoring the supporting muscles of the scapula, which Hatcher considered diagnostic of the 1st dorsal vertebra. Modified from Hatcher (1903: plate I).

However none of these definitions is wholly satisfactory. Rib fusion is not a reliable criterion, as Hatcher (1903: 8) notes: “there are in our collections of sauropods, skeletons of other dinosaurs fully adult but, with the posterior cervical, bearing free cervical ribs articulating by both tubercular and capitular facets as do the ribs of the dorsal region”. As one example, the Mamenchisaurus hochuanensis holotype CCG V 20401 has free ribs on its last three cervicals (Young & Zhao, 1972; personal observation). It may be argued that those last three cervicals are really dorsals, but while a case could possibly be made for the last one, the rib shafts of the previous two are much too horizontally oriented to sustain such an interpretation.

Ventral centrum shape is highly variable between different sauropod taxa—and, as noted above, centra are prone to distortion, though this is probably less prevalent in the relatively robust ventral part of the centrum than in the more fragile cotyle.

The presence of a scapular support facet on the lateral face of the CPRL is difficult to detect, and evidently not trusted by Hatcher himself as he noted of C15 of Diplodocus CM 84 that “the superior of the anterior blades of the horizontal lamina has its external surface somewhat expanded and rugose. It no doubt served as a support for the muscular attachment of the heavy scapular arch” (Hatcher, 1901: 25).

While the parapophysis migrates dorsally across the centrum to the arch in successive vertebrae of all sauropods, this migration typically does not commence until after the first dorsal, making it useless to determining the location of the junction.

The genetic definition is obviously useless for fossil organisms.

In practice, most workers seem to use a combination of multiple criteria, often not explicitly specified. For example, in considering the dorsal count of Barosaurus, McIntosh (2005: 45) writes “The eighth presacral [counting forward from the sacrum] is unquestionably a dorsal and the tenth a cervical”, but does not explain why he makes these designations. He continues, “But what is the ninth? At first glance it certainly appears to be a cervical—the parapophysis projects from the very bottom of the centrum well below the pleurocoel. However […] neither rib is co-ossified to the parapophyses and diapophyses as in the cervical vertebrae anterior to it. Largely for this reason, I have concluded that it is the first dorsal.” Here, McIntosh is making a judgement on the contradictory evidence of the rib-fusion and parapophysis-location criteria, while not using ventral centrum shape (perhaps because the ventral view of the centrum is not available) or scapular-support facets (perhaps due to preservational problems making it impossible to assess). This example is instructive, as it illustrates pervasive difficulties when dealing with bones as large, fragile and difficult to manipulate as sauropod presacrals: very few such bones can be inspected from all cardinal directions, and as noted above distortion and damage are ubiquitous.

Perhaps, then, the best we can hope for in identifying the cervicodorsal junction is to use a combination of criteria according to what is available for study in the specimen in question, but to be explicit about which criteria are used. As McIntosh’s example demonstrates, this identification is important, as it determines the number of cervical vertebrae deemed to belong to a neck: even in those very rare cases when a presacral vertebral sequence is complete and undistorted, it is still to some degree a matter of judgement how many cervical vertebra constitute the neck.

(In some older papers (e.g. Migeod, 1931), a “shoulder vertebra” is referenced, an intermediate between the last cervical and the first dorsal, but this terminology is not used in modern literature. However, introducing this concept does not aid the quest for consensus over how the junction is located: the issue simply becomes the problem of locating the shoulder vertebra, rather than locating the first dorsal.)

Discussion

Taphonomic factors

All of the problems with sauropod neck preservation arise from the nature of the animals and the general procedures of taphonomy.

First, sauropods are big. This is a recipe for incompleteness of preservation: small skeletons are more easily destroyed by taphonomic processes, but if they survive are more easily preserved whole, while large skeletons less rarely survive intact (Brocklehurst et al., 2012). It is no accident that the most completely preserved sauropods are small individuals such as CM 11338, the cow-sized juvenile Camarasaurus lentus described by Gilmore (1925). For an organism to be fossilised, it is necessary for the carcass to be swiftly buried in mud, ash or some other substrate. This can happen relatively easily to small animals, such as the many finely preserved small theropods from the Yixian Formation in China, but is much less possible with a large animal (Mannion, 2010: 284).

Cleary et al. (2015: 528 and figure 6) showed that medium-sized ichthyosaurs preserve more completely than either small or large individuals, but since these are aquatic animals their preservational context is not applicable to the case of sauropods. Brown et al. (2012) found that in the Dinosaur Park Formation, “large-bodied” dinosaurs preserved more completely than smaller ones, but their sample contained no sauropods, their threshold for “large” was only 60 kg, and the largest animals included were 4.5-tonne hadrosaurs. It may be that if the methods of Brown et al. (2012) were used to analyse the sauropod-bearing Morrison or Tendaguru formations, the result would be similar to those of Cleary et al. (2015), with medium-sized animals having the most complete preservation. At a larger scale, Cashmore et al. (2020: 963) found only a weak, and weakly significant, correlation between sauropod body mass and specimen completeness (R2 = 0.03; p = 0.04), using the SCM2 skeletal completeness metric of Mannion & Upchurch (2010). However, analysis of their figure 5A, plotting log body mass against log completeness shows a rapid falling away in completeness above body masses with log 10—presumably natural log, representing about 22 tonnes.

It is also possible that the light construction of highly pneumatic cervical vertebrae would have rendered them particularly prone to water transport, disarticulating and scrambling the necks of even some otherwise adequately preserved specimens.

Except in truly exceptional circumstances, sediments simply are not deposited quickly enough in terrestrial environments to cover a 25 m, 30 tonne animal with a light neck skeleton before it is broken apart by scavenging, decay and water transport. Fossilisation of the very largest sauropods tends to produce even more fragmentary remains. In light of this, it is not surprising that the very longest sauropod necks are usually known from particularly inadequate specimens. The longest neck for which we have direct evidence is that of the diplodocid Supersaurus, possibly 15 m long, but the only cervical material of the largest specimen is a single 1.4 m cervical (BYU 9024, formerly BYU 5003; Jensen, 1985, 1987). Similarly, the giant basal titanosauriform Sauroposeidon probably had a neck about 11 m long, but the only definite material belonging to it is a sequence of three and a half cervicals from the middle of the neck (OMNH 53062; Wedel, Cifelli & Sanders, 2000). The longest known titanosaur necks are probably those of Patagotitan, Puertasaurus and Dreadnoughtus, all at around 9–10 m, but the cervical material from which they are known is meagre: only three vertebrae in the Patagotitan holotype MPEF-PV 3400, of which the longest is 120 cm long (supplementary information to Carballido et al., 2007); a single 118 cm Puertasaurus vertebra, MPM-PV 10002 (Novas et al., 2005); and a single 113 cm vertebra of Dreadnoughtus MPM-PV 1156 (Lacovara et al., 2014).

Secondly, even when complete sauropod skeletons are preserved, or at least complete necks, distortion of the preserved cervical vertebrae is almost inevitable because of their uniquely fragile construction. As in modern birds, the cervical vertebrae were lightened by extensive pneumatisation, so that they were more air than bone (Taylor & Wedel, 2013a: figure 4), with the air-space proportion typically in the region of 60–70% and sometimes reaching as high as 89% (Taylor & Wedel, 2013a: table 2; Wedel, 2005: figure 7.4C). While this construction enabled the vertebrae to withstand great stresses for a given mass of bone, it nevertheless left them prone to crushing, shearing and torsion when removed from their protective layer of soft tissue. For highly pneumatized cervicals in particular, the chance of the shape surviving through taphonomy, fossilisation and subsequent deformation would be tiny.

Possible mitigations

Some information about sauropods necks that is not directly observable can be inferred from phylogenetic bracketing: the observation that if two close outgroups to an organism of interest both have a feature, then the null hypothesis is that the organism of interest inherited that feature from the common ancestor (see Witmer, 1995). For example, the neck of Sauroposeidon proteles is known only from a sequence of four vertebrae (Wedel, Cifelli & Sanders, 2000). But as a brachiosaurid it is bracketed by Giraffatitan brancai, which is thought on reasonable evidence (see above) to have had 13 cervicals, and Camarasaurus lentus, which is known from complete specimens to have had 12 cervicals (Gilmore, 1925: 367). It is therefore parsimonious on this basis to conclude that the most recent common ancestor of Giraffatitan and Camarasaurus probably had 12–13 cervical vertebrae, and that Sauroposeidon proteles likewise also had 12–13 cervicals.

While this level of inference is better than no information, the present example illustrates three problems with the use of phylogenetic bracketing for sauropod morphology. First, due to incomplete knowledge of most sauropods, it is often necessary to move some way down the tree from the organism of interest before reaching specimens for which morphology is sufficiently known. In this example, while Giraffatitan is probably close to Sauroposeidon, Camarasaurus is some way basal to it—but more closely related brachiosaurids such as Abydosaurus and Europasaurus, while possessing good cervical material, do not have anything close to a complete neck, and so cannot be used in bracketing. As a result, the Giraffatitan–Camarasaurus bracket used in this example is broader than we would wish. Second, the phylogenetic context in which we interpret brackets is not always well established. In the present example, I have been following Wedel, Cifelli & Sanders (2000) referral of Sauroposeidon to the clade Brachiosauridae; but if D’Emic & Foreman (2012) are correct in their assessment that Sauroposeidon is in fact a somphospondylian, then it would be more appropriate to bracket it between Giraffatitan brancai and Euhelopus zdanskyi—which has 17 cervical vertebrae (Wiman, 1929: 7), suggesting a significantly higher cervical count of 13–17 for Sauroposeidon. Third, in the case of sauropods, the morphology of the bracketing taxa is often not as solidly established as we might wish. In this case, while the cervical counts of Camarasarus lentus and Euhelopus zdanskyi are well established, that of Giraffatitan brancai is less so, as documented above, rendering the cervical-count ranges for Sauroposeidon even more uncertain. In conclusion, while phylogenetic bracketing remains a useful tool, it must be used with care.

The problem of distortion in individual vertebrae can be addressed to some extent through retrodeformation (e.g. Schlager et al., 2018). This term encompasses a suite of techniques in which mathematical processes are applied to virtual models of fossils in an attempt to restore them to the shape they had before their deformation in matrix. Various methods have been proposed in the literature, largely based on an assumption of symmetry. Most of the early methods were designed for 2D transformations on morphologically simple invertebrate fossils (e.g. Cooper, 1990), but more recent developments have yielded methods more capable of handling complex 3D shapes (e.g. the Single Axis Method of Kazhdan et al., 2009). Application of these techniques to sauropod vertebrae has so far been limited, but Tschopp, Russo & Dzemski, 2013 conducted experiments on cervical vertebrae of the diplodocid Kaatedocus siberi using an artificially deformed model of a dodo cervical as a control. They found that while the retrodeformation techniques they tested were capable of restoring symmetry to a deformed vertebra, they often did so at the expense of artificially making the vertebra shorter, broader or thinner than it should be. Surprisingly, they also found that applying the same process twice to a vertebra first made it more robust than the original, then more slender. At this stage it seems that more sophisticated techniques, preserving more aspects of the 3D geometry, will be necessary before retrodeformed sauropod cervicals can be taken at face value.

But this does not mean that other strategies cannot help in dealing with distorted material. For example, Hedrick & Dodson (2013) used 3D geometric morphometric techniques to better understand individual and taphonomic variation in a set of 30 skulls representing three nominal species of Psittacosaurus. They showed that the apparent morphological variation between these “species” actually represent the results of different taphonomic processes, and that all 30 skulls in fact represent members of the same species. It may be possible to use similar techniques on sauropod vertebrae, although modifications would be required to deal with the additional complexity introduced by serial variation in vertebral morphology.

One thing that is apparent in all this is the necessity of careful documentation on how specimens were found, in what states of articulation, in what kinds of matrix, and under what strains: for example, Angielczyk & Sheets (2007) use strain measurements to quantify deformation. These descriptions should where possible be accompanied by quarry maps, and ideally by photos and even 3D models of the excavation—something that is increasingly possible given the unlimited capacity for online supplementary information in many journals. Such additional documentation may not materially increase our understanding of the specimens but does at least make it more explicit what we do and do not know, yielding rigidly defined areas of doubt and uncertainty. Similarly, when fossils are restored, it should be done in such a way that it is apparent what parts of the restored specimen are real bone and which are reconstructed. Happily, modern palaeontologists are much better in all these respects than our predecessors were. Some “golden age” palaeontologists were concerned primarily with what would constitute a spectacular museum mount: it is for this reason that O. C. Marsh had YPM 1980, the holotype of Brontosaurus excelsus, so enthusiastically “restored” that it is now impossible to determine which parts of the cervical vertebrae are real (personal observation; Barbour, 1890). But these days are mostly behind us, and nearly all modern specimens are prepared in a form that accurately conveys what was actually preserved.

Implications

Both the incompleteness and distortion of sauropod necks have grave consequences for our ability to reason about sauropods. As noted above, the very small sample of complete necks makes it quite impossible to perform meaningful statistical analyses. This incomplete information also impedes our ability to understand the evolution of sauropod necks, making it difficult to determine the plesiomorphic cervical morphology and vertebral counts at the bases of clades. Similarly, the frequent, unpredictable and sometimes dramatic distortion of what vertebrae we do have renders mechanical analysis of neutral poses and ranges of motion extremely problematic. For vertebrae small and robust enough to be manipulated by hand, this can be readily observed in physical space (Fig. 13). There is no reason to think that computer modelling of vertebrae and their articulations should yield models any more informative than the distorted fossils that they are based on.

On a more positive note, the lack of complete necks does not mean that we are without information. For many sauropods that lack complete necks, enough vertebrae are preserved with enough fidelity that we can have a good idea how morphology varies between anterior, middle and posterior cervicals, even if precise identification of the vertebrae is not possible. Crucially, this degree of completeness suffices for the majority of characters to be scored in phylogenetic analyses: apart from a few characters specific to the atlas or axis, most such characters pertain only to anterior, middle or posterior cervicals.

Conclusion

What does it all mean? Only this: we don’t know as much as we may assume we do. We don’t even know how many cervical vertebrae well-known sauropods such as Diplodocus and Giraffatitan had. We don’t have complete necks for either of these sauropods, nor for almost any others. Even those we do have are in some cases badly crushed (e.g. Mamenchisaurus hochuanensis). We are woefully short of sauropod necks.

As scientists, we must carefully avoid blithely asserting factoids such as “Diplodocus had 15 cervicals and Giraffatitan only 13”. We simply don’t know know whether this is true. Evidence supports it as a hypothesis—these numbers are certainly the best guesses for the taxa in question—but a hypothesis is all it is. Hypotheses of neck posture and flexibility should be held even more lightly, since they are based on inferences drawn from distorted elements whose true shapes we may never know.

None of this is necessarily disastrous, so long as we properly acknowledge the degree of uncertainty that afflicts our work. Problems arise when studies such as that of Stevens & Parrish (1999) draw apparently firm conclusions about sauropod neck posture based on specimens that are deficient in respects not acknowledged in the text, lending the results a veneer of definitiveness that they do not merit.

An earlier version of this manuscript was reviewed in 2015 by Paul Barrett, Paul Upchurch and Jeffrey Wilson; the present version was reviewed by Oliver Rauhut, Paul Upchurch (again) and Veronica Diez-Diaz. This final version is much stronger for having been updated in response to their comments. I also thank Brandon Hedrick for his editorial handing of the manuscript, and my wife, Fiona, for motivating me to finally get the revisions done on this long overdue paper.

After the first version of this manuscript was published as a preprint, numerous people reminded me of additional complete and near-complete sauropod necks. These include (in alphabetical order) Lee Braithwaite, John D’Angelo, Vahe Demirjian, Oliver Demuth, Daniel Gonçalves, Stephen Gunnell, Rutger Jansma, Matt Lamanna, Stefan Reiss, Emmanuel Tschopp, Justin Tweet and Paul Upchurch.

I thank Jorge O. Calvo, Paul C. Sereno, Leonardo S. Filippi, Rodolfo A. Coria, Kristian Remes, Ralf Kosma, Jose I. Canudo and Stephen F. Poropat for permission to cite personal communications.

I thank Mathew J. Wedel for his observations on the phylogenetic distribution of complete and near-complete necks, and for providing the juvenile alligator neck illustrated in Fig. 8. Matt Cobley provided the ostrich neck illustrated in Fig. 7. Andrew Moore supplied photographs of the neck of the Camarasaurus lentus specimen USNM 13786. Daniela Schwarz provided high-resolution scans of Werner Janensch’s site maps of “Quarry S”, where important Giraffatitan material was found. Gerhardt Maier provided the translation of Janensch (1950). This translation, and others that I referred to, are freely available from the Polyglot Paleontologist web-site at http://paleoglot.org/.

This paper is based in part on three blog-posts from Sauropod Vertebra Picture of the Week (Taylor, 2011, 2013, 2014), and also on part of a talk at the Symposium on Vertebrate Palaeontology and Comparative Anatomy (Taylor & Wedel, 2011).

Institutional abbreviations

AODF Australian Age of Dinosaurs Fossil, Winton (Australia)

BYU Brigham Young University, Provo, Utah (USA)

CCG V Chengdu College of Geology, Chengdu (China), vertebrate collection

CM Carnegie Museum of Natural History, Pittsburgh, Pennsylvania (USA)

GCP Grupo Cultural Paleontológico de Elche, Museo Paleontológico de Elche (Spain)

IVPP Institute of Vertebrate Paleontology and Paleoanthropology, Chinese Academy of Sciences, Beijing (China)

MACN Museo Argentino de Ciencias Naturales ‘Bernardino Rivadavia’, Buenos Aires (Argentina)

MAU-Pv Museo Argentino Urquiza, Rincón de los Sauces, Neuquén (Argentina), vertebrate palaeontology collection

MB.R see MfN

MCT Collection of the Earth Science Museum of the National Department of Mineral Production, Rio de Janeiro (Brazil)

MfN Museum für Naturkunde Berlin, Berlin (Germany): collection numbers for fossil reptiles: MB.R.nnnn

MNBH Musée National Boubou Hama, Niamey (Republic of Niger)

MOZ-Pv Museo Provincial de Ciencias Naturales “Dr. Prof. Juan A. Olsacher”, Zapala (Argentina), vertebrate palaeontology collection

MPEF Museo Paleontológico Egidio Feruglio, Trelew (Argentina): collection numbers for fossil vertebrates: MPEF PV

MPM Museo Padre Molina, Río Gallegos, Santa Cruz (Argentina): collection numbers for fossil vertebrates: MPM PV

MUCPv Museo de Geologia y Paleontologia de la Universidad Nacional del Comahue, Neuquén (Argentina), vertebrae palaeontology collection

OMNH Sam Noble Oklahoma Museum of Natural History, Norman, Oklahoma (USA)

PMU Paleontological Museum, Uppsala, Sweden

QJGPM Qijiang Petrified Wood and Dinosaur Footprint National Geological Park Museum, Chongqing (China)

SMA Sauriermuseum Aathal (Switzerland)

SSV Shanshan Geological Museum, Shanshan (China)

USNM National Museum of Natural History, Smithsonian Institution, Washington, D.C. (USA)

YPM Yale Peabody Museum, New Haven, Connecticut (USA)

ZDM Zigong Dinosaur Museum, Zigong, Sichuan (China)

Additional Information and Declarations

Competing Interests

Author Contributions

Data Availability

The author declares that he has no competing interests.

Michael P. Taylor conceived and designed the experiments, performed the experiments, analyzed the data, prepared figures and/or tables, authored or reviewed drafts of the paper, and approved the final draft.

The following information was supplied regarding data availability:

There is no raw data beyond what is contained in the article.

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
