# Peer review of "Almost all known sauropod necks are incomplete and distorted"

_PeerJ, doi:10.7717/peerj.12810_

## Round 0.1 · original submission · Major Revisions

Dear Dr. Taylor,

Thank you for your submission to PeerJ. After receiving comments from three reviewers and reading through the paper myself, I find that it would be suitable for publication in PeerJ following major revisions.

As noted by the first and second reviewer, the paper is centered on a fact, that sauropod necks, and in fact fossils, are distorted and incomplete. Certainly true, and while not a novel idea to paleobiologists, it is certainly one that is useful to be reminded of and I found this paper to be a thorough review of the issue concerning sauropod necks. That said, I do agree with reviewer 2 that there are times where the paper reads as though paleobiologists are not aware of issues of completion or taphonomic shape change. I would tone these instances down, especially in the places where reviewer 2 suggests.

I would also strongly echo reviewer 1's point that the discussion needs a 'where do we go from here' section. As of now, the paper points out an issue and then kind of fizzles a bit. However, especially recently there has been quite a bit of work done on better understanding the impacts of taphonomy on shape (including some done by me). Here are a few references that might be useful to help deal with distortion.

Hedrick, B.P., Schachner, E.R., Rivera, G., Dodson, P. and Pierce, S.E., 2019. The effects of skeletal asymmetry on interpreting biologic variation and taphonomy in the fossil record. Paleobiology, 45(1), pp.154-166.

Kammerer CF, Deutsch M, Lungmus JK, Angielczyk KD. Effects of taphonomic deformation on geometric morphometric analysis of fossils: a study using the dicynodont Diictodon feliceps (Therapsida, Anomodontia). PeerJ. 2020 Oct 7;8:e9925.

Wynd, B.M., Uyeda, J.C. and Nesbitt, S.J., 2021. Including distorted specimens in allometric studies: linear mixed models account for deformation. Integrative Organismal Biology, 3(1), p.obab017.

Lefebvre, R., Allain, R., Houssaye, A. and Cornette, R., 2020. Disentangling biological variability and taphonomy: shape analysis of the limb long bones of the sauropodomorph dinosaur Plateosaurus. PeerJ, 8, p.e9359.

Hedrick, B.P. and Dodson, P., 2013. Lujiatun psittacosaurids: understanding individual and taphonomic variation using 3D geometric morphometrics. PLoS One, 8(8), p.e69265.

Baert, M., Burns, M.E. and Currie, P.J., 2014. Quantitative diagenetic analyses of Edmontosaurus regalis (Dinosauria: Hadrosauridae) postcranial elements from the Danek Bonebed, Upper Cretaceous Horseshoe Canyon Formation, Edmonton, Alberta, Canada: implications for allometric studies of fossil organisms. Canadian Journal of Earth Sciences, 51(11), pp.1007-1016.

I suppose the issue of incompleteness can only be solved by sending people out to dig more! However, I am most interested in the ideas presented by reviewer 2 (how do removing characters likely impacted by taphonomy affect trees? how does distortion impact functional analyses?). I consider these beyond the scope of this paper, but I definitely think that they should be discussed and suggestions should be made.

Thank you for your submission to PeerJ. Please let me know if you have any questions and I will be happy to answer them.

Best,

Brandon P. Hedrick, Ph.D.

·

Basic reporting

no comment

Experimental design

no comment

Validity of the findings

no comment

Additional comments

I liked reading this article: In a time of ever more sophisticated methods, it is good to be made aware of the general imperfections of the fossil record - your method might be extremely exact and sophisticated, but if the basic data is insufficient, your results can still be highly questionable. I thus recommend the article for publication. As this is mainly a manuscript reminding us of this simple fact, I have little to add, and have made most of my comments on the annotated manuscript, which is attached. I would only add a few further, more general comments here.

- This article makes it clear how important mode of preservation and taphonomy is for our understanding of fossil organisms. Even if a skeleton was not found articulated, the exact mode of preservation can make a difference to our understanding of its original anatomy. Important aspects might be the degree of disarticulation, the transport or sorting of elements, general modes of compaction or distortion, and, very basic, whether remains of other animals have been found in the same locality or not. All of this information will have an impact on how reliable a reconstruction of such aspects as neck length might be seen. It might be worth to point this out somewhere in the text: report not only what of an animal has been preserved, but also what has not been preserved, what other animals might be present, and how the remains were found.

- As for the materials of Giraffatitan (skeletons SI and SII), I would be less critical than the author concerning the articulated series of SII. From the comments by Janensch and additional field sketches (one of which was reproduced by Heinrich 1999: fig. 18) it seems clear that most of the presacral vertebral column of this individual was indeed found in articulation, with only one vertebrae having been slightly rotated from its original position (but without disturbing the sequence of vertebrae). Thus, the sequence of vertebrae seems to be at least complete as far as it was preserved (with no additional elements missing) - whether the first preserved vertebra was C3 (as argued by Janensch) or might have been another number, cannot of course be said on the basis of these considerations alone.

- Identification of cervicodorsal transition: It might be worth to see how this is in other archosaurs, as similar problems exist, for example, in theropods. Here, the most reliable identification of the first dorsal vertebra is the form of the rib - cervical ribs are parallel to the vertebral column, whereas dorsal ribes stick away from it, and this is usually used to identify this transition in cases where you have complete, articulated skeletons. Other possibilities for identification exist as well, though, and it might be good to reference some of these problems here.

- In the discussion it might be good to also show ways towards possible solutions. Do you think digital retrodeformations might help with deformed elements? And if so, what would be the criteria for retrodeformation? What is your view of the evolution of cervical vertebral number in sauropods, in the light of your results from looking at how much we know about complete series? Might phylogenetic bracketing help in decisions about how many vertebrae there were?
One thing that I think is clear from your manuscript is that careful documentation of the taphonomy of specimens in the original osteological description can indeed help to judge whether a given number of cervical vertebrae is realistic, or should be seen with caution. The same argument could be made for species for which more than one specimen is known - if you can find reliable criteria for identifying vertebral number in partial articulated specimens, this can then be used to overlap different individuals to reach a reliable vertebral count (Does this apply to any of the cases dealt with in the manuscript? Given, for example, that multiple specimens of Diplodocus are known, is it really impossible to say with certainty that Diplodocus had 15 cervicals, or can we get to this conclusion by overlapping several partial, articulated skeletons?). I would suggest to add these points to the discussion to improve anatomical description and reasoning for giving vertebral counts.

·

Basic reporting

This is a clearly written and well presented paper, with high-quality figures. I would like to compliment the author on the detail of his scholarship, with extensive examination of the relevant literature. I found a few typos, grammatical errors, and other minor problems,listed below –

Lines 22-23 Abstract – ‘deposited quickly enough to cover a giant carcass on land; and distortion of presacral vertebrae is common due their lightweight hollow construction. This ubiquitous incompleteness’

‘due their’ should be ‘due to their’

Line 44 – ‘…The consequent incompleteness of known sauropods’ necks, and the ubiquitous…’

This should be ‘sauropod necks’

Lines 227-228 – ‘…Mamenchisauridae — although it is unfortunate many of these specimens are not well well described in English:’

Repetition of ‘well’

Lines 600 onwards – discussion of sauropod body size and preservation. This section would benefit from incorporation of the findings of Cashmore et al. 2020, Palaeontology.

Lines 638-641 - For highly pneumatized cervicals in particular, the chance of the shape surviving through taphonomy, fossilisation and subsequent deformation would be tiny, as is also the case with the vertebrae of highly pneumatic fossil birds.’

This statement about fossil birds could do with a supporting reference.

Lines 668-676 - ‘…we may never know. None of this is necessarily disastrous, so long as we properly acknowledge the degree of uncertainty that afflicts our work. Problems arise when studies such as that of Stevens and Parrish (1999) draw apparently firm conclusions about sauropod neck posture based on specimens that are deficient in respects not acknowledged in the text, lending the results a veneer of definitiveness that they do not merit. It might be argued that the venue of this study (Science) is too constrained for space to allow a proper discussion of degrees of uncertainty; this may be true, but is really only an argument that such venues are not suitable for the publication of rigorous scientific work.’

First, I think that suggesting that Science might not be a suitable venue for rigourous scientific work could cause offence. Second, in any case, the issue of limited space in short format journals has been greatly alleviated by the ability to publish extensive online supplements. I think the author should remove or greatly modified the point being made here.

Experimental design

I have two fundamental, and I fear fatal, problems with this paper. First, and most importantly, I think it attempts to address a problem that does not really exist. It sets up a strawman with regard to the need to tell researchers that sauropod next are less complete than we previously thought. However, I would argue that we are well aware of these issues and that the current paper does not provide convincing evidence that there is a problem with the way we are doing things now. To be clear, I am not saying that the incompleteness of sauropod necks is not a problem – it definitely is. What I’m saying is that there is little value in a paper whose main message is to tell us what we already know and take into account (as far as we can) when looking at neck function and phylogenetic characters. The author does not provide a convincing case for why the details he gives need to be presented to his fellow researchers – he would need to document substantial cases of current or very recent work that has gone badly wrong because of an unawareness of the issues – generally this justification is missing from the paper (see below(. It may well be that members of the public etc are misled by the level of certainty put across in some scientific papers – but the author has already published three blogs and a preprint on this issue, I would imagine that that is sufficient. The scientific community working on these issues does not need to be reminded of the general importance of understanding the limitations on the data we use

The second problem is that the paper is rather qualitative, with the possible exception of the introduction of some indices relating to aspect ratios (which is novel). There are a number of ways in which the paper could become more quantitative and statistical, and I suggest some of these below. Nevertheless, without a meaningful research problem to address, the addition of such analyses would be pointless at the current time. I justify these views below by quoting portions of the paper and giving my responses.

Lines 38-39 – ‘…rudimentary measurements to be made – for example, neck length or shoulder height. In fact, the problem is more significant than is generally realised. It is not just that we do not have large…’

The utility of this paper hinges largely on the claim that the problem is greater than is generally realised as stated above. However, there are no references to back this up (at leat from the last 20 years), and no analyses to demonstrate that palaeobiologists are making the mistake of ignoring the problems with sauropod neck preservation. Rather, those of us working on these animals are very well aware of the problems already, down to the individual specimen level, but we use the data we have with caveats (a good example is the set of assumptions made by Bates et al 2016 when they needed to estimate sauropod neck lengths and volumes). If the relevant workers are already aware of the problems, then what is the value of publishing a paper that tells us what we already know? None. The author must show that a reasonable number of recent studies have made Significant errors by assuming greater completeness of necks than is warranted.

Bates, K. T., Mannion, P. D., Falkingham, P. L., Brusatte, S. L., Hutchinson, J. R., Otero, A., Sellers, W. I., Sullivan, C., Stevens, K. A. and Allen, V. (2016). Temporal and phylogenetic evolution of the sauropod dinosaur body plan. Royal Society Open Science, 3, 150636.

Similarly –

Lines 256-257 – ‘Several well-known sauropod specimens are often thought of as having complete, undamaged necks, but in each case the truth is less clear. I now discuss three important specimens.’

And

Line 658 – ‘What does it all mean? Only this: we don’t know as much as we may assume we do.’

Again, there are no references cited in the above quotes to back up these claims. Who has claimed these necks are undamaged and complete? I bet that in most cases, palaeobiologists have described these as ‘nearly’ or ‘virtually’ complete, or a vague ‘well preserved’ – rather than really stating that these are truly undamaged and complete even down to the atlas etc (when is a sauropod ever undamaged in the strict sense?]. To suggest that palaeobiologists working on sauropods are unaware of these issues or are making false assumptions about completeness/preservation is, to my mind, setting up a straw man that is then easily knocked down in this paper.

In fact, I would argue that even the evidence presented by the author supports my point about the attitude of palaeobiologists. The examples given of researchers perhaps being over-confident are from very old literature – Hatcher 1901 etc. Other contemporaries such as Holland were clearly sceptical about the number and completeness of the cervicals of Diplodocus, and even Hatcher’s work includes some caveats. Where is the evidence that even these historic researchers, let alone modern ones, are working on the basis that sauropod necks are complete and undamaged?

The nearest we get to modern researchers making a false assumption about neck completeness and getting it wrong is given in the section on Apatosaurus. Here, Stevens and parrish 1999 assumed a complete and well preserved neck for their biomechanical modelling, even though the last three cervicals were badly crushed and heavily restored. So, modern researchers can be over confident at times., However, interestingly this problem was spotted by Upchurch 2000 (both references cited in the current paper). So, clearly researchers are already on the lookout for such unwarranted assumptions, and either address them in their own work, or criticise others when they overstep the bounds of what represents a justifiable simplification/assumption/extrapolation made for the purposes of making progress.

Around lines 490, the author also cites the Elongation Index as a good example of where ignoring distortion could cause major errors in phylogenetic data, taxonomic comparisons or functional analyses. I agree. However, the elongation index has been largely replaced by the Average Elongation Index. Whereas the former is based on centrum length divided by height ()and is therefore very prone to problems caused by distortion), the latter uses centrum length divided by the average of the height and width of the cotyle. It is conceivable that, as a centrum is crushed vertically so that its height is artificially reduced, is width is also artificially increased. Therefore, AEI might be less prone to problems caused by crushing. We cannot be sure of this, but the author’s point would be enhanced if he looked at AEIs. Indeed, a comparison between AEI and EI could be instructive, as would a comparison of AEIs across sauropod necks and extant taxa. I encourage the author to go down this route in the future.

Lines 658-667 – ‘…We don’t even know how many cervical vertebrae well-known sauropods such as Diplodocus and Giraffatitan had. We don’t have complete necks for either of these sauropods, nor for almost any others. Even those we do have are in some cases badly crushed (e.g. Mamenchisaurus hochuanensis). We are woefully short of sauropod necks. As scientists, we must carefully avoid blithely asserting factoids such as “Diplodocus had 15 cervicals and Giraffatitan only 13”. We simply don’t know know whether this is true. Evidence supports it as a hypothesis – these numbers are certainly the best guesses for the taxa in question – but a hypothesis is all it is. Hypotheses of neck posture and flexibility should be held even more lightly, since they are based on inferences drawn from distorted elements who true shapes


I think the author overstates the case here. Clearly such statements are hypotheses and should be regarded as such. But I would argue that they are already regarded as hypotheses. seWe all, as professionals, talk to each other in a shorthand which to the outsider might sound like we are pronouncing facts when we actually mean hypotheses. This is done for brevity, not because we have assumed that hypotheses are facts. For example, it would become rather tedious if every paper on sauropods had to say things like ‘Giraffatitan is thought to have 13 cervicals, although there are several factors that mean caution is required’. In any case, I think my colleagues are fairly cautious, and I’ve often seen ‘estimated or ‘?’ placed before such numbers. No doubt the literature contains some sloppy or imprecise phrasing, but there is no need to write a paper based on this, unless the author can show that such imprecision is causing major errors in our understanding of the animals.

For me, the strongest and most useful part of the paper is the review of how the cervicodorsal junction is identified in sauropods. certainly, this is a problem, and the author is right that many modern workers still state cervical and dorsal numbers without clearly explaining why they have drawn the line between them. However, this section needs further work. For example, if we are uncertain how to draw the line between cervicals and dorsals, how can we cite evidence that critiques some of the criteria used to define the cervicodorsal junction? A good example is that of Mamenchisaurus hochuanensis, cited in this paper. the author points out that posterior cervicals can have free rather than fused ribs, and mentions that the last three cervicals of M. hochuanensis have free cervical ribs. That’s correct, but this depends on the view that the cervicodorsal junction occurs after presacral 19 – if we assume that there were actually just 16 cervicals, then the free ribs would only occur on dorsals, and the valid status of this criterion would be restored. Such argumentation would, of course, then contradict other criteria such as parapophysis position etc. – but this example serves to demonstrate the difficulties in adequately analysing the strengths and weaknesses of different criteria for definition of the cervicodorsal junction, since the evidence for or against them in part depends on first knowing where the cervicodorsal junction is. Moreover, for me, this discussion of the cervicodorsal junction is incomplete. Many other workers have discussed this issue – e.g., see Wilson and UPchurch 2009 – and there are other criteria that could be considered (e.g. number of vertebrae anterior to the first appearance of the hyposphene). In short, this section has potential, but in its current form I do not think it stands up as a publishable unit in its own right.

Validity of the findings

The findings are valid insofar as they provide evidence that sauropod necks are incomplete and discuss these issues in detail. The evidence that incompleteness and distortion are widespread is correct. However, as noted above, I do not find that the research question is a valid one for a scientific publication. Documenting a problem is only useful if fellow researchers are not aware of it and are making invalid assumptions. Of course, given the available sauropod necks, we do make some assumptions and approximations in order to use data for functional and phylogenetic work – what the author needs to do is demonstrate that the conclusions from such work have been significantly compromised buy such assumptions and approximations because of a lack of awareness of sauropod neck incompleteness and distortion – but he does not really do this.

Also –

Lines 189-192 -• ‘…MOZ-Pv1232, the holotype of Lavocatisaurus agrioensis, described by Canudo et al. (2018). This is complete from C1-C11. Canudo’s guess is that this is complete neck (Jose I. Canudo, pers. comm, 2021), but the specimen doesn’t demand that conclusion and no known eusauropod has fewer than 12 cervicals.’

This claim about no known eusauropod having fewer than 12 cervicals is bold and problematic given that this paper also argues, quite rightly, for difficulties in estimating cervical number because of the problem of determining exactly where the cervicodorsal junction lies. For example, in Amargasaurus there are just 22 presacrals – it is hard to tell how many of these are dorsals and how many cervicals - but in my personal observation of the specimen I suspect that there might just be 11. So, a more cautious and nuanced statement is needed here.

Additional comments

I reviewed the previous version of this paper. The author has certainly attempted to take my criticisms, and those of the other reviewers, into account. Certainly the discussion of various necks is much more comprehensive now. However, the fundamental problem has not been addressed, because in my view it cannot be addressed. As I have said above, telling fellow researchers that a problem exists, when those researchers are already aware of that problem, does not represent a meaningful Advanced in our knowledge. I recommend that the author should focus on a more quantitative assessment of Completeness and distortion, and then use these data to look for potentially interesting taxonomic and/or taphonomic patterns. This might actually make a useful and novel contribution. For example, rather than the qualitative assessments of neck completeness in the current paper, perhaps the author could look at completeness from the point of view of the completeness metrics developed by Mannion and Upchurch and subsequently used by many others. This could be combined with the AEI study noted above. Finally, it is one thing to show that distortion affects the proportions of cervicals, but the author needs to show that this makes a difference to phylogenetic and functional studies – e.g., buy showing that variation in proportions from one cervical to the next means that a taxon scores a 0 and a 1 for a given character. He might also form collaborations with other workers to look at how distortion might affect functional interpretations and whether this is adequately handled By current sensitivity analyses.

·

Basic reporting

This manuscript is a really interesting (critical) review on more classical bibliography, gathered together with actual references and discussions with other researchers.

By providing in-deep analyses and discussions on the preserved sauropod remains and their descriptions, this work opens our minds, questioning the validity of the term "completeness" when speaking to articulated/associated fossil remains. This should serve as basis for re-studing the other sauropod "complete" necks, but also when describing other skeletal material.

The table and figures are informative and complement with the text of the manuscript.

I have no further comments or suggestions, and recommend this manuscript for its publication as it is.

Experimental design

No comment.

Validity of the findings

No comment.

---

## Round 0.2 · Minor Revisions

Dear Dr. Taylor,

Thank you for your careful revisions and thoughtful changes in light of reviewer comments. I think this is an excellent paper and will be very useful to researchers (especially new researchers) interested in studying sauropods. There are a lot of thoughtful points that you bring up. On a final read through, I found a few points that will require a bit more revision. Once this is done, I will be happy to move this paper along to the proof stage.

When you resubmit, please submit a tracked changes version and clean version of the manuscript. Please let me know if you have any questions.

Best,

Brandon P. Hedrick, Ph.D.




Line 120: In the catalogue of complete necks, I understand knowing that you have the first cervical, but is it certain in all these that you have the last? Maybe add in a sentence about that before listing the specimens in this section? I see that it has its own section below, but it would be good to mention here too.

Line 260: Can you include an access date here for PD?

Line 419–420: Saying you plan to redescribe it seems odd. I would delete this.

Line 524: aEI? On 520 you say aEI rather than eEI. Which is correct?

Line 530: Can you say what you mean by whole vertebra distortion? Do you mean the cranial part of a vertebra is deformed in one way and the caudal part of the same vertebra is deformed differently?

Line 539: I always find this interesting. Papers refer to specimens as being ‘relatively undistorted’. However, what is distorted to one researcher is not necessarily distorted to another.

Line 590: And likely subject to distortion?

Line 700: Why is Europasaurus underlined?

Line 713: Just a thought, but it would be interesting to look at bird cervical count and see how variable they are within orders and then within families. I would imagine that it would be tied more to ecology than phylogeny, but I would think that data would be available for analysis. That might have some bearing on this discussion.

Line 727: I think this is very important. Retrodeformation as it is currently done is often just symmetrization. However, even making the vertebrae symmetric would only remove the asymmetric component of variation. I’m not sure what could be done to remove perfectly symmetric distortion without knowing what the original shape was. I discuss this at length in Hedrick et al. 2019; Paleobiology https://round-lake.dustinice.workers.dev:443/https/doi.org/10.1017/pab.2018.42. I think it’s a really important topic and appreciate your bringing it up.

Line 741: Check out Angielczyk and Sheets (2007; https://round-lake.dustinice.workers.dev:443/https/doi.org/10.1666/06007.1). They use strain to assess deformation and I think you might add a sentence about it and cite them.

Line 743: I’m not sure about the word unlimited here.

Line 748: ‘much much’ is too colloquial

Table 1: Include a column ‘units’ so readers know what the units are even though they are consistent within an animal.

---

## Round 0.3 · accepted · Accept

Dear Dr. Taylor,

Thanks for handling this so quickly and being so careful with both reviewer comments and my comments through the process. This is a very nice paper and I am excited to see it published. I have moved it on to the next stage. Please let me know if you have any questions and I will be happy to answer them.

Best,

Brandon P. Hedrick, Ph.D.